# Test performance of lateral flow rapid antigen tests for COVID-19 in Welsh adult care home staff using routine surveillance data

**Craig Hogg** [1‡]*, **Sian Boots**[1‡], **Daniel Howorth**[1], **Christopher Williams**[1], **Margaret Heginbothom**[1], **Jane Salmon**[1], **Robin Howe**[2]

**1** Communicable Disease Surveillance Centre, Public Health Wales, Cardiff, Wales, United Kingdom,
**2** Public Health Wales Microbiology, Cardiff, Wales, United Kingdom

‡ CH and SB are joint senior authors to this work.
* Craig.Hogg@wales.nhs.uk

## Abstract

### Background

Lateral flow tests (LFTs) have been used to screen for SARS-CoV2 in Wales since January 2021. Between May and August 2021, adult care home staff policy was for biweekly Innova LFT and weekly Polymerase Chain Reaction (PCR) testing while asymptomatic. We estimated test performance of LFTs conducted in adult care home staff using PCR tests as a reference standard.

### Methods

Test results from surveillance data were matched by individual where both LFT and PCR were taken on the same day. We calculated sensitivity, specificity, positive and negative predictive values, and agreement using Matthew's correlation coefficient. Ct values of positive PCR results were compared by matched LFT result. Analysis was conducted using R v4.1.3.

### Results

We analysed 115,593 test pairs, 499 (0.43%) of which were PCR positive. Median age was 48 (IQR: 22) and 85.00% of the study population were female. Test result agreement was 99.59% (95%CI 99.55–99.63; MCC: 0.38, p<0.001). Sensitivity and specificity were 25.65% (95%CI 22.02–29.67) and 99.91% (95%CI 99.89–99.93), respectively. PPV was 55.90% (95%CI 49.42–62.17) and NPV was 99.68% (95%CI 99.64–99.71). Crude Ct values were significantly lower in positive PCR tests matched to a positive LFT compared to a negative LFT.

### Conclusions

Specificity and negative predictive value were high in an asymptomatic population of care home staff indicating this test is an effective tool for identifying cases of SARS-CoV-2 infection during periods of high prevalence where transmission is likely, due to the presence of

**Data Availability Statement:** All relevant data are within the paper and its Supporting information files.

**Funding:** The author(s) received no specific funding for this work.

**Competing interests:** The authors have declared that no competing interests exist.

high viral loads. Positive predictive value results are lower than existing literature yet should be considered in light of the asymptomatic study population and low prevalence (under 1%) at the time most of these tests were conducted. Performance improved at times of higher prevalence during the study. These results suggest that whilst lateral flow tests are effective for identifying SARS-COV-2 infections with high viral loads, they are not effective at identifying cases with a low viral load. When an LFT provides a negative result, false negatives should be considered and additional diagnostic tests performed.

## Background

On the 11[th] March 2020 the World Health Organisation declared a coronavirus disease-19 (COVID-19) pandemic as a result of a newly identified Severe Acute Respiratory Syndrome-related coronavirus (SARS-CoV-2) [1, 2]. Worldwide epidemiological surveillance systems to monitor COVID-19 have developed rapidly as has the development of rapid non-invasive antigen lateral flow tests (LFTs) designed to detect SARS-COV-2. Whilst polymerase chain reaction tests (PCRs) are able to pick up lower levels of SARS-COV-2 than LFTs, rapid antigen testing can provide results quickly and cost effectively [3, 4]. They can therefore be a useful early detection tool for COVID-19 case clusters in asymptomatic individuals [5]. However, there has been much debate as to whether LFTs exhibit adequate performance to meet the needs of a mass-surveillance and screening system [6].

In January 2021, the Welsh Government launched a COVID-19 testing strategy that included the use of LFTs to aid in preventing the transmission of COVID-19 across Wales [7]. A part of this strategy involved the use of LFTs in closed settings such as care homes to reduce the risk of transmission from staff to care home residents. In these closed settings, asymptomatic staff members were advised to undertake Innova LFTs twice or more per week along with a weekly Polymerase Chain Reaction (PCR) test [8].

In January 2022, policy change removed the requirement to confirm a positive LFT results with a PCR test [9]. In April 2022, policy identified LFTs as the primary testing route for the symptomatic diagnosis of COVID-19. These testing policies and their subsequent changes (as is applicable for all testing policies) require careful evaluation, particularly so with the rapid changes to testing policies implemented throughout the pandemic. Interpretation of this evaluation will be affected by restrictions and management of cases deemed to be 'false positive' [10], and the use of LFTs as a routine screening tool [11].

With the data generated as a result of the implementation of the Testing Strategy for Wales, we aim in this paper to measure the sensitivity and specificity of LFTs against PCRs for those tests taken in Adult Care Home settings in Wales during the period 1[st] May 2021 – 31[st] August 2021.

## Methods

### Study population

Individuals were included in the study population where their postcode of residence was in Wales, and they were of working age (between 18 and 67 years old at time of first test within the timeframe). Both the LFT result and PCR result must be positive or negative (not void), and conducted between 1[st] May 2021 and 31[st] August 2021. From this study population, we then only included individuals whose LFT record was associated with an adult care home, and

who identified as a member of staff in the care home. Following guidance, symptomatic individuals should book a PCR test in the first instance, therefore this study population is presumed to be primarily asymptomatic.

## Data sources

Lateral flow testing activity data for Wales was sourced from the UK Government Lateral Flow Testing Portal, collated by NHS Digital, and supplied to Public Health Wales (PHW) via a data feed and a set of SQL views from Digital Health and Care Wales (DHCW).

PCR testing data was sourced from NHS and lighthouse laboratories across Wales and supplied to PHW through a data feed from DHCW.

A list of adult care home unique organisation numbers (UONs) and UON groups in line with the Testing Strategy for Wales was provided to PHW and DHCW by the Welsh Government.

All data was collected as part of routine surveillance and management of COVID-19. Patient identifiable information was anonymised when necessary and held securely on the NHS Wales Network.

## Test matching

Lateral flow test records and PCR test records were matched on a 1:1 basis based on a unique reference code where both LFT and PCR were taken on the same day. Where an individual conducted multiple tests of the same type on the same day, the earliest lateral flow with a valid result was matched with the earliest PCR with a valid result conducted on that day; all other tests on that day were excluded. Subsequent LFTs and PCRs were matched in the same way in order to capture repeat testing on different days.

## Lateral flow assay

Lateral flow assay tests were taken by the individual using Innova test kits [12]. Presuming guidance was followed correctly, self-administered combined nose and throat swabs were sampled and interpreted at 30 minutes. Results were then self-reported in conjunction with contact and demographic information to an online portal hosted by the UK Government (https://www.gov.uk/report-covid19-result).

## PCR assay

RT-PCR was used as the 'reference standard' in this analysis. No classification was recorded regarding whether the swab was conducted by the individual or a trained professional. Combined nose and throat swabs or dry throat swabs were sampled and sent to a variety of NHS Wales laboratories and lighthouse laboratories. RT-PCR was then conducted. Results were graded and authorised by trained laboratory staff according to manufacturer definitions and standard operating procedures [13]. Sample processing equipment specification was likely to vary by time and test processing location due to the use of multiple PCR platforms across NHS Wales and lighthouse laboratories, and therefore was not identical in every test. Ct values for positive PCR results were not available for stratified analysis.

## Statistical analysis

Specificity, sensitivity, and predictive values were calculated using the RT-PCR results as reference standard. Where applicable, 95% Confidence Intervals for proportions were calculated using the Wilson method. Matthew's correlation coefficient (MCC) was used to evaluate the

**Table 1. Summary of characteristics of test records which contributed toward the study population.**

| | |
|---|---|
| Total *N* (Valid LFTs with a linked valid PCR) | 115,593 |
| Positive PCR (% (*n*)) | 0.43% (499) |
| Positive LFT (% (*n*)) | 0.20% (229) |
| Sex (%F (*n*))[a] | 85.00% (98,741) |
| Age (Median (IQR)) | 48 (22) |

[a] Sex was unknown or unspecified in 116 records

level of agreement between tests. Ct values were extracted where available for PCR results and were compared between LFT positive and LFT negative groups, using Mann-Whitney to test for significance. Statistical analyses were performed using R v4.1.0.

## Ethics

Public Health Wales was established under The Public Health Wales National Health Service Trust (Establishment) Order 2009. Its functions include the provision and management of a range of public health, health protection, healthcare improvement, health advisory, child protection and microbiological laboratory services and services relating to the surveillance, prevention and control of communicable diseases. This work falls under the establishment order and therefore no further NHS research permissions were required. Data were held and processed under Public Health Wales information governance arrangements in compliance with the Data Protection Act, Caldicott Principles and Public Health Wales guidance on the release of small numbers. All methods were performed in accordance with the relevant guidelines and regulations. No data identifying protected characteristics of an individual were released outside Public Health Wales. Data used in the study was anonymised prior to analysis. By choosing to take a test for COVID-19 via the NHS Test and Trace programme, of which all data in this study falls under, implied consent was obtained for the use of data collected for surveillance and research purposes [14].

## Results

### Characteristics of the study population

Characteristics of the selected study population (n = 115,593) are shown in Table 1. These demographics are shown at test level and include repeated testing by individuals.

### Test results

In these 115,593 linked test records, 0.43% of PCR results were positive (n = 499) and 99.57% were negative (n = 115,094), whilst 0.20% of LFT results were Positive (n = 229) and 99.80% were negative (n = 115,364) (Table 2).

**Table 2. Summary of Innova LFT results compared to PCR.**

| | Innova Lateral Flow | | |
|---|---|---|---|
| **RT-PCR** | Positive | Negative | TOTAL |
| Positive | 128 | 371 | 499 |
| Negative | 101 | 114,993 | 115,094 |
| TOTAL | 229 | 115,364 | 115,593 |

**Table 3. Test performance measures calculated for Innova lateral flow device in asymptomatic care home staff.**

| Measure | Estimate (95% CI) |
|---|---|
| Sensitivity (95%CI) | 25.65% (22.02, 29.67) |
| Specificity (95%CI) | 99.91% (99.89, 99.93) |
| Positive Predictive Value (95%CI) | 55.90% (49.42, 62.17) |
| Negative Predictive Value (95%CI) | 99.68% (99.64, 99.71) |
| Agreement (95%CI; MCC) | 99.59% (99.55, 99.63; 0.38) |

Agreement between groups was calculated to be 99.59% (MCC: 0.38; 95%CI 99.55, 99.63). Of the 115,593 tests in the study population, 472 matched tests did not agree. A total of 78.60% (n = 371) of these cases had a negative LFT followed by a Positive PCR ("False Negatives") whilst 21.40% (n = 101) had a positive LFT followed by a negative PCR ("False Positives").

## Test performance

Using this data, overall sensitivity of the lateral flow devices was 25.65% (95%CI 22.02–29.67) and specificity was 99.91% (95%CI 99.89, 99.93). Across the entire study period, Negative Predictive Value (NPV) was estimated to be 99.68% (95%CI 99.64, 99.71), whilst Positive Predictive Value (PPV) was estimated to be 55.90% (95%CI 49.42, 62.17) (Table 3). PPV however varied over time due to changing prevalence during the study period, ranging from a minimum (non-zero) estimate of 22.22% to a maximum of 85.19% (Fig 1) [15].

## PCR Ct values

Ct values were available for 51 (10.2%) of positive PCR results included in the study. We tested the distribution of Ct results and found them not to be normally distributed (Shapiro-Wilk:

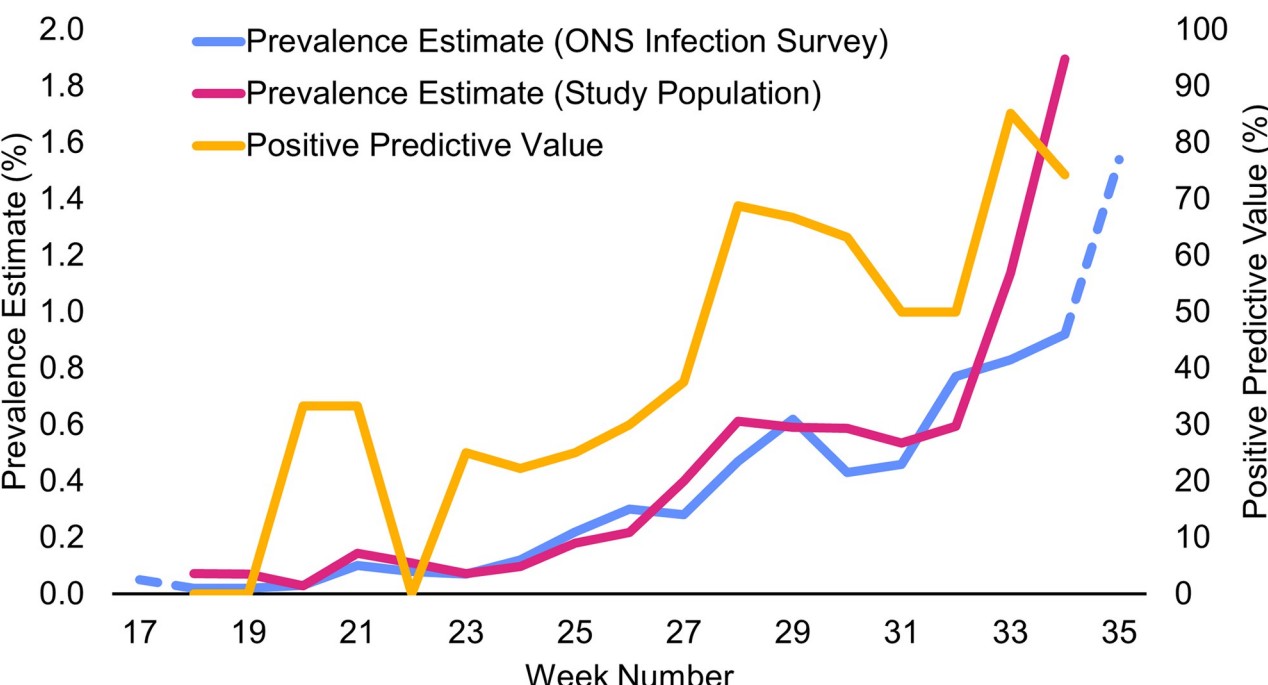

**Fig 1. Prevalence estimate comparison (left axis) and Positive Predictive Value (right axis) by week.** Data excluded for weeks 17 and 35 due to incomplete weekly data.

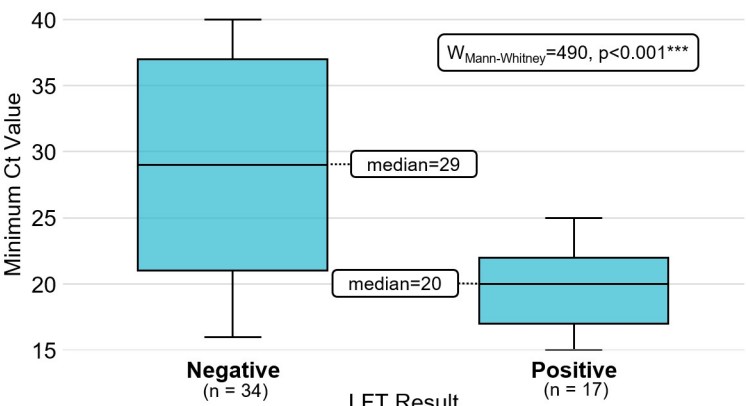

**Fig 2. Minimum target Ct value, by matched LFT result.** Labels show median of each group and statistical difference between groups.

W = 0.904, p < 0.001***). Control target values were available for less than half of these results (45.1%) so we did not proceed to normalisation of minimum Ct values. Wilcoxon rank sum test yielded a p value < 0.001, suggesting a significant difference in crude minimum Ct values between matched LFT result groups (Fig 2).

## Discussion

We believe our study at time of publication to be the first study of lateral flow test performance using self-reported results from the UK Government online portal demonstrating real-world test performance estimates. The study has identified that Innova lateral flow testing kits supplied as part of the Testing Strategy for Wales yielded a high specificity reaching above 99%, indicating that lateral flow tests could be an effective tool to rule in SARS-CoV-2 infection where the test result is positive due to the very low false positive rate during a time of low prevalence. We also believe that the use of LFTs during this time period promoted testing in an asymptomatic population, and therefore likely identified many cases which would otherwise have gone unidentified. Our study's low PPV results can be attributed to the asymptomatic study population and low prevalence (under 1%) of SARS-COV-2 during the study's analytical time period. PPV increases notably with increased prevalence, however sensitivity analysis, which is unaffected by prevalence, yielded a lower sensitivity than expected with an overall estimate of 25.65%. The low sensitivity of lateral flow tests identified in this study therefore does not support the use of lateral flow tests as a reliable diagnosis platform as the test cannot reliably rule out infection, particularly during times of low prevalence where NPV will be reduced. Our study's sensitivity results contrasts with the results of a systematic review by Mistry *et al*. [16], which calculated a pooled sensitivity estimate of 78.68% in Innova test kits.

Previous studies have demonstrated lower sensitivity estimates in self-swabbing samples compared to those in studies where the test was undertaken by a trained professional [17]. However, a study by Amoo et al. [17] found that self-sampling for qPCR analysis was reliable and yielded a high agreement percentage compared with samples collected by healthcare workers. A likely major factor contributing to the low results found in our study is that the testing programme is targeted towards asymptomatic individuals, which has been shown to yield lower sensitivity estimates than a symptomatic population [18]. We also hope that the large sample size of this study will mitigate bias introduced by the effects of possible poor sampling technique.

Viral load is also a possible contributing factor to the likelihood of detection in lateral flow devices [19], however evidence is limited to support a difference in viral load between symptomatic and asymptomatic cases [20, 21]. A further limitation is that less than half of available Ct values had control values which were able to be used to analyse normalised Ct results. Whilst preliminary analysis of the available Ct values suggested a high proportion of LFT negatives had lower Ct values indicative of higher viral load, a more robust approach would be required to draw any meaningful conclusion from such analysis.

Research has shown a person may shed SARS-COV-2 for up 17 days, however live virus was only found within 9 days [22, 23]. Therefore, as a result of the high sensitivity of PCRs, they could yield a positive result for a longer period of time compared to a LFT which only detect higher viral loads, creating a narrow time period where both results would give a positive result [23]. There is also variability in sensitivity and specificity of PCR tests which may differ depending on the manufacturer of the assay or machine used. Differing factors may therefore influence test performance characteristics due to variance in the PCR as 'reference standard' test.

Due to the repeated testing in our study population, it is possible that previously PCR positive individuals may continue to test positive by PCR where public testing guidance would advise against repeated testing in this post-infectious period [24, 25]. As Lateral Flow Tests would be unlikely to identify infection in this period [23], this aspect of the study population's testing regime may negatively impact the sensitivity estimates in this analysis. There may also be implications to the calculation of prevalence in this scenario, and therefore may partially account for the Positive Predictive Value of 55.90%. However, comparison with ONS Coronavirus Infection Survey [13, 26] showed similar prevalence to the overall Welsh population at the time the tests of our study were conducted (Fig 1), further supporting the reliability of the PPV estimate given in this paper.

Further validation of occupational data would be beneficial on a wider scale, as initial data exploration showed inconsistencies in reported job roles and reasonable demographic information. Regardless, the linkage of data to a care-home via unique organisation number suggests that the tests were likely to have been taken in a domestic care setting and would therefore be conducted in accordance with guidance.

We propose the study contributes further to the justification of the use of lateral flow testing at times of high prevalence and pre-test probability due to the high testing specificity and PPV at times of increased prevalence. It is also reassuring to consider the large number of asymptomatic cases which would have otherwise gone unidentified in this population. The low sensitivity identified is concerning, however the sensitivity results should be considered in light of prevalence at the time the study's testing results were reported, alongside the bias toward asymptomatic cases in the study population leading to low numbers in some subgroups.

The time period in which this study was conducted was chosen to reflect a time where regular asymptomatic testing was promoted, however the wide scope of testing at the time will undoubtably include pauci-symptomatic individuals and those with atypical symptoms. Whilst this limits the generalisability of these findings to a strictly asymptomatic population, we believe this study remains reflective of testing behaviour at the time, particularly with regard to social behaviour and individual decision making following results. In light of this, the results of this study highlight the potential scale of false positive results and the consequences this may introduce when applied to our study population of care home staff. There was notable impact on staffing levels in care homes during this period [27], not only impacting patient care but increasing burden of work for other healthy staff. A high number of false positives will therefore needlessly increase this burden. Future testing policy should therefore take these findings into account and it is hoped that

the scale of this problem will allow policymakers to make more informed decisions when balancing this risk against the benefit of regular testing leading to reduced transmission where positive results are, in fact, reflective of SARS-CoV-2 infection. In addition to this, the impact of false negatives must be considered due to the risk of increased transmission if a false negative result is used to justify work or social attendance. The low sensitivity in this analysis indicates that this could be a substantial factor to consider when implementing similar testing policies in future.

Public messaging must continue to promote the reliability of lateral flow tests whilst making clear the limited ability of the test to rule out SARS-CoV-2 infection [6, 28]. The self-reporting mechanism in place throughout the first two years of the COVID-19 pandemic offered a unique and extensive system by which data from symptomatic and asymptomatic testing streams could be collected separately and collated with regard to the limitations of each system. Wider qualitative study is needed to understand attitudes towards rapid testing, and the reliability of reported results in light of policy change and individual decision making.

## Supporting information

**S1 Dataset.**
(XLSX)

## Acknowledgments

We would like to thank all staff involved in the curation and supply of data flows used in COVID-19 surveillance in Wales, particularly Gareth John from Digital Health Care Wales, and Helen Clayton and Laura Dexter from Public Health Wales for their work in the development and upkeep of COVID-19 testing data used in this analysis. Thanks also to Laia Fina for epidemiological advice in the early stages of this project. Finally, we would like to extend our gratitude to all laboratory staff working in both NHS Wales and Lighthouse labs for their hard work processing COVID-19 tests throughout the pandemic. The author(s) read and approved the final manuscript.

## Author Contributions

**Conceptualization:** Craig Hogg, Sian Boots, Margaret Heginbothom, Jane Salmon, Robin Howe.

**Data curation:** Craig Hogg, Sian Boots.

**Formal analysis:** Craig Hogg, Sian Boots, Daniel Howorth, Christopher Williams.

**Investigation:** Craig Hogg, Sian Boots.

**Methodology:** Craig Hogg, Sian Boots.

**Project administration:** Craig Hogg, Sian Boots.

**Supervision:** Margaret Heginbothom, Jane Salmon.

**Visualization:** Craig Hogg, Sian Boots.

**Writing – original draft:** Craig Hogg, Sian Boots, Daniel Howorth.

**Writing – review & editing:** Craig Hogg, Sian Boots, Daniel Howorth, Christopher Williams, Margaret Heginbothom, Jane Salmon, Robin Howe.

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
