## [Decision Letter · Decision Letter 0]

25 Apr 2023

PONE-D-23-02428Test performance of lateral flow rapid antigen tests for COVID-19 in Welsh adult care home staff using routine surveillance dataPLOS ONE

Dear Dr. Hogg,

Thank you for submitting your manuscript to PLOS ONE. After careful consideration, we feel that it has merit but does not fully meet PLOS ONE’s publication criteria as it currently stands. Therefore, we invite you to submit a revised version of the manuscript that addresses the points raised during the review process.

This manuscript is quite interesting, especially due to the fact that it works with real-world data and information obtained in epidemiological surveillance activities, managing to bring insights that contribute to scientific knowledge and to the practice of health services. I forward below the comments of the reviewers and the next steps to be taken.

Note that reviewer #1 comments are noted in the body of this email. Reviewer #2 comments are indicated in the PDF document in the form of annotations that appear in the attached file.

Please reply to each comment from both in the reply email.

We look forward to receiving your revised manuscript.

Kind regards,

André Ricardo Ribas Freitas

Academic Editor

PLOS ONE

Journal Requirements:

2. Please ensure that you have specified (1) whether consent was informed and (2) what type you obtained (for instance, written or verbal, and if verbal, how it was documented and witnessed). If your study included minors, state whether you obtained consent from parents or guardians. If the need for consent was waived by the ethics committee, please include this information

3. We noted in your submission details that a portion of your manuscript may have been presented or published elsewhere. "Analysis contained in this work was presented as a poster at the European Scientific Conference for Applied Infectious Disease Epidemiology in Stockholm, November 2022. Poster as presented is attached as a file." Please clarify whether this [conference proceeding or publication] was peer-reviewed and formally published. If this work was previously peer-reviewed and published, in the cover letter please provide the reason that this work does not constitute dual publication and should be included in the current manuscript.

Reviewers' comments:

Reviewer's Responses to Questions

**Comments to the Author**

1. Is the manuscript technically sound, and do the data support the conclusions?

Reviewer #1: Yes

Reviewer #2: Yes

2. Has the statistical analysis been performed appropriately and rigorously? 

Reviewer #1: Yes

Reviewer #2: Yes

3. Have the authors made all data underlying the findings in their manuscript fully available?

Reviewer #1: Yes

Reviewer #2: Yes

4. Is the manuscript presented in an intelligible fashion and written in standard English?

Reviewer #1: Yes

Reviewer #2: Yes

5. Review Comments to the Author

Reviewer #1: Thank you for the opportunity to review this evaluation of the use of lateral flow tests for SARS-CoV-2 by care home staff in Wales. The study uses nationally collected records reflecting real world regular use of LFTs at a time when weekly PCR testing was also recommended. Although its results are likely to be a good reflection of test performance characteristics it is not a prospectively designed test accuracy study and is therefore somewhat limited in the extent to which underlying reasons for observed results can be investigated. It is nevertheless an interesting addition to the evidence base for LFT use during the pandemic. I have made a few comments below that the authors may wish to consider.

Pg 3, line 52-59, Abstract > Conclusions. The first line states that LFTs are effective for ruling in infection and last line states that they are an unsatisfactory tool. I understand that the mitigating factor is prevalence however this does not come across clearly enough in the text and read as an apparent contradiction in terms. Some rewording might help.

Pg 4, line 84-85, Background. “Rapid changes in policy increase the need to evaluate…” is a bit weak, suggest could put more emphasis on the fact that any testing policy (mass or otherwise) should always be evaluated carefully prior to roll-out.

Pg 5, line 99-101, Methods > Study population. This is the only place where the authors acknowledge the assumption that the study population is primarily asymptomatic; this could be returned to in the Discussion as potential limitation of the study. Individuals will not always adhere to national testing policies, and an ‘assumed asymptomatic’ group will include pauci-symptomatic and those with atypical symptoms. On the plus side, although symptom status cannot be properly investigate the study remains a real world reflection of testing behaviour, at least for those who bothered tor register their test results.

Pg 6, line 125, line 135. Please replace ‘gold’ standard with ‘reference standard’ which is now the more commonly accepted term; ‘gold’ alludes to ‘perfection’ when there is no such thing as a perfect reference test.

Pg 6, line 128-132. PCR assay – no information is provided about the RT-PCR assays used in the study; ideally at least some details should be given even in an appendix

Pg 7, line 137-138 Statistical analysis. Why calculate agreement between tests when the interest in test accuracy, particularly in this scenario with many thousands of participants, low prevalence and very high specificity? I do not think it adds any value to the paper. Also suggest that comparing Ct values with significance values is a mistake given the apparently numerous RT-PCR assays used, inability to normalise values and the small proportion of PCR results for which Ct values were available.

Pg 10, line 189-190. Suggestion of a significant difference in crude minimum Ct values – again suggest that adding statistical significance here is not particularly helpful. Could also point to the fact that a high proportion of LFT negatives had Ct values in the lower range (higher viral load) which would lead to a different interpretation of data than that inferred by a ‘statistically significant difference’

Pg 10 Discussion. Includes frequent use of ‘this’ near the beginning of sentences (e.g. line 204 ‘This value…’, line 206 ‘Due to this, this analysis…’, line 209 ‘This contrasts…’. It is a bit of a bugbear but it really helps the reader if you could be more specific about exactly what ‘this’ is. The first paragraph of the Discussion, in particular, would be much easier to follow if the concepts presented were more clearly explained. Similarly to the Abstract, the test is promoted as both an effective tool and ‘not a reliable diagnostic platform’ and could be confusing for those not fully familiar with how prevalence affects accuracy.

Pg 11-12 Discussion. Much is made here of possible factors that could have affected observed results with statements made that further research is needed to explain findings. In fact much of this research has already been done, e.g. in regard to symptoms, viral load and timing of tests, and what this study does is reflect all of those factors without having the available data to properly investigate those. I suggest this section is considerably shortened, highlighting these contributing factors and referring appropriately to available literature that is already out there supporting this. The whole paragraph on self swabbing (line 246-266) could really be dropped altogether as it is completely confusing and tries to make way too much of results regarding agreement which are not helpful in the first place.

Pg 11-12 Discussion. What is missing from the Discussion is any attempt to consider the potential harms and benefits from the LFT testing policy, e.g. false positive results leading to care home staff unnecessarily staying away from work with possible economic and social consequences, and those with false negative results who may easily be infections or becoming infectious remaining at work and potentially leading to increased transmission of infection.

Reviewer #2: The paper Hogg et al describes the performance of lateral flow tests (LFT) to detect SARS-CoV-2 as a screening tool among a majority asymptomatic population (adult care home staff). Analyzing surveillance data that captured both LFT and PCR test results, the authors showed that the LFT used for screening (Innova) had a high specificity but low sensitivity, and that sensitivity increases based on increase in COVID-19 prevalence, and conclude that LFT is most useful to rule in COVID-19 and especially useful at times of high prevalence. The authors provided sound analyses and conclusions. My minor comments in the text revolve around providing more clarity on the testing policy at the time in the methods section, as well as shifting some of the discussion section to improve flow. I've attached my comments into the PDF. Many thanks to the authors for their great work, especially in showing how surveillance and testing data can be used to assess diagnostic performance based on different use cases.

6. PLOS authors have the option to publish the peer review history of their article (what does this mean?). If published, this will include your full peer review and any attached files.

Reviewer #1: **Yes: **Jacqueline Dinnes

Reviewer #2: No

---

## [Author Response · Author response to Decision Letter 0]

19 Jun 2023

Reviewer comments

We would like to thank both reviewers for your kind comments, we are glad that the benefits of using real world surveillance data in this study were clear throughout the paper. We understand the limitations of this study introduce difficulty in further investigating the findings of the study, however we hope we have adequately described the extent of these limitations and further work which could improve this were this methodology to be repeated. All of the below comments have allowed us to improve the quality and clarity of our paper and we are very grateful for the insight your collaboration has given us.

Reviewer 1 Comments

Thank you for the opportunity to review this evaluation of the use of lateral flow tests for SARS-CoV-2 by care home staff in Wales. The study uses nationally collected records reflecting real world regular use of LFTs at a time when weekly PCR testing was also recommended. Although its results are likely to be a good reflection of test performance characteristics it is not a prospectively designed test accuracy study and is therefore somewhat limited in the extent to which underlying reasons for observed results can be investigated. It is nevertheless an interesting addition to the evidence base for LFT use during the pandemic. I have made a few comments below that the authors may wish to consider.

1. Pg 3, line 52-59, Abstract > Conclusions. The first line states that LFTs are effective for ruling in infection and last line states that they are an unsatisfactory tool. I understand that the mitigating factor is prevalence however this does not come across clearly enough in the text and read as an apparent contradiction in terms. Some rewording might help.

We agree with this point and have amended our abstract to clarify our conclusions.

2. Pg 4, line 84-85, Background. “Rapid changes in policy increase the need to evaluate…” is a bit weak, suggest could put more emphasis on the fact that any testing policy (mass or otherwise) should always be evaluated carefully prior to roll-out. 

We agree this wording should be stronger and have amended this sentence to reflect this comment. 

3. Pg 5, line 99-101, Methods > Study population. This is the only place where the authors acknowledge the assumption that the study population is primarily asymptomatic; this could be returned to in the Discussion as potential limitation of the study. Individuals will not always adhere to national testing policies, and an ‘assumed asymptomatic’ group will include pauci-symptomatic and those with atypical symptoms. On the plus side, although symptom status cannot be properly investigate the study remains a real world reflection of testing behaviour, at least for those who bothered to register their test results.

Thank you for noting this as we strongly agree this is an important point to consider and so have addressed this point in the second to last paragraph of the discussion. We hope this added paragraph offers a clear explanation of how this is a limitation of the study. We have also added further detail regarding the impact of real world decision making in an asymptomatic group which we hope will add value to this point.

4. Pg 6, line 125, line 135. Please replace ‘gold’ standard with ‘reference standard’ which is now the more commonly accepted term; ‘gold’ alludes to ‘perfection’ when there is no such thing as a perfect reference test.

We have replaced all use of ‘gold standard’ with the term ‘reference standard’.

5. Pg 6, line 128-132. PCR assay – no information is provided about the RT-PCR assays used in the study; ideally at least some details should be given even in an appendix

We would liked to have provided more information on the assays used, however as this data was collated using routine surveillance data we have very little detail on the assays used in lighthouse laboratories particularly. In addition to this, at least 13 different PCR platforms were used in NHS Wales laboratories. Due to the wide range of platforms used and the lack of complete data, we have tried to include more clarity on the fact that samples were sent to a wide range of laboratories and tested using multiple PCR platforms.

6. Pg 7, line 137-138 Statistical analysis. Why calculate agreement between tests when the interest in test accuracy, particularly in this scenario with many thousands of participants, low prevalence and very high specificity? I do not think it adds any value to the paper. 

We feel that the agreement provides a useful overview of test concordance, and as such feel that there is some value added by including this figure. However we agree that all included figures may not be needed and so have removed the p value calculated using Fisher’s exact test as we do not feel it adds further value than that provided by the agreement figure in combination with MCC. 

7. Also suggest that comparing Ct values with significance values is a mistake given the apparently numerous RT-PCR assays used, inability to normalise values and the small proportion of PCR results for which Ct values were available.

Although sample size was less than we would have liked for this piece of analysis, we still feel there is merit in the comparison of these groups. Grading of positives and low level positives was conducted using Ct values which were not normalised, and therefore these figures are somewhat reflective of the results which would have guided decision making were an individual to receive a PCR result. We feel that the significant difference in these figures supports our analysis in showing that a positive LFT result may be indicative of a higher viral load, and that LFT results may be influenced by the point an individual is at in the cycle of infection as it relates to viral load. Whilst the difference in median Ct value can be clearly seen from the figures, we feel that including a statistical test adds robustness to this observation, and that doing so does not take away any value from this analysis. 

8. Pg 10, line 189-190. Suggestion of a significant difference in crude minimum Ct values – again suggest that adding statistical significance here is not particularly helpful. Could also point to the fact that a high proportion of LFT negatives had Ct values in the lower range (higher viral load) which would lead to a different interpretation of data than that inferred by a ‘statistically significant difference’

We hope we have addressed this comment in the point above.

9. Pg 10 Discussion. Includes frequent use of ‘this’ near the beginning of sentences (e.g. line 204 ‘This value…’, line 206 ‘Due to this, this analysis…’, line 209 ‘This contrasts…’. It is a bit of a bugbear but it really helps the reader if you could be more specific about exactly what ‘this’ is. The first paragraph of the Discussion, in particular, would be much easier to follow if the concepts presented were more clearly explained. Similarly to the Abstract, the test is promoted as both an effective tool and ‘not a reliable diagnostic platform’ and could be confusing for those not fully familiar with how prevalence affects accuracy.

We agree with this statement fully. We have clarified sentences where possible to reduce use of the word ‘this. We have also amended the first paragraph of the discussion and our abstract to clarify our conclusions. 

10. Pg 11-12 Discussion. Much is made here of possible factors that could have affected observed results with statements made that further research is needed to explain findings. In fact much of this research has already been done, e.g. in regard to symptoms, viral load and timing of tests, and what this study does is reflect all of those factors without having the available data to properly investigate those. I suggest this section is considerably shortened, highlighting these contributing factors and referring appropriately to available literature that is already out there supporting this. The whole paragraph on self swabbing (line 246-266) could really be dropped altogether as it is completely confusing and tries to make way too much of results regarding agreement which are not helpful in the first place.

Upon rereading this paragraph, we agree with these points and have removed the paragraph on self swabbing. We have adjusted the discussion to more clearly explain how this work reflects these factors and added a considerably shorter comment regarding self swabbing to the second paragraph of the discussion. We feel the point regarding self-swabbing is now better supported here without overinflating it’s influence and without relying on results regarding agreement. 

11. Pg 11-12 Discussion. What is missing from the Discussion is any attempt to consider the potential harms and benefits from the LFT testing policy, e.g. false positive results leading to care home staff unnecessarily staying away from work with possible economic and social consequences, and those with false negative results who may easily be infections or becoming infectious remaining at work and potentially leading to increased transmission of infection.

Agree this is a very important point to make, and so have added further discussion regarding harms and benefits of this testing policy in the penultimate paragraph of the discussion. 

Reviewer 2 Comments

The paper Hogg et al describes the performance of lateral flow tests (LFT) to detect SARS-CoV-2 as a screening tool among a majority asymptomatic population (adult care home staff). Analyzing surveillance data that captured both LFT and PCR test results, the authors showed that the LFT used for screening (Innova) had a high specificity but low sensitivity, and that sensitivity increases based on increase in COVID-19 prevalence, and conclude that LFT is most useful to rule in COVID-19 and especially useful at times of high prevalence. The authors provided sound analyses and conclusions. My minor comments in the text revolve around providing more clarity on the testing policy at the time in the methods section, as well as shifting some of the discussion section to improve flow. I've attached my comments into the PDF. Many thanks to the authors for their great work, especially in showing how surveillance and testing data can be used to assess diagnostic performance based on different use cases.

1. Pg 4, line 79, Background. “closed settings”: such as?

We have added further clarification “such as care homes” here.

2. Pg 4, line 81, Background. Suggest also mentioning that PCR was done once a week as well (it’s mentioned in the abstract but not the full text).

We have added clarification to include this at the end of this sentence.

3. Pg 4, line 82, Background. “LFD” LFT?

Apologies, corrected to LFT.

4. Pg 6, line 114, Methods > Test Matching. Add “based on”

We agree this is clearer and have implemented this change

5. Pg 6, line 117, Methods > Test Matching. Unclear as written that the analysis included repeat testing (which was mentioned in the results)

Reworded to clarify that other tests matched for a person on the same day were excluded and that we captured repeat testing on different days in this analysis.

6. Pg 6, line 120, Methods > Lateral Flow Assay. Recommend adding a reference to the instruction for use of this test

We have added reference to a guidance document for this purpose, however note that while this document is still available at time of writing, it is not easily found on DHSC web pages and many other related documents have been withdrawn due to policy change. 

7. Pg 6, line 123, Methods > Lateral Flow Assay. Possible to add link to the online portal?

We have added a link to this in brackets within the text.

8. Pg 6, line 128, Methods > PCR Assay. Possible to add where testing was done (e.g. at a central laboratory? At the facility?

We have added further explanation to clarify this and highlighted that the samples were sent to a range of laboratories for processing.

9. Pg 10, line 202, Discussion. I would not say “PPV results are much lower than existing literature” since it’s clear that PPV is affected by prevalence. I would instead suggest saying that PPV is low because of the study population and low prevalence of COVID-19 during this time period.

Thank you for this comment. We agree that this change adds clarity given the point we were trying to make and have implemented this change.

10. Pg 11, line 214, Discussion. The premise of the paragraph of self-swabbing and contributing factor of asymptomatic population don’t make sense together. The first couple of sentences make more sense with the paragraph on self-sampling below. However, highlighting that the study population was among asymptomatic individuals is important, and should be kept, and perhaps incorporated into the concluding paragraph instead.

In response to comments by reviewer 1 we have edited this section and added text from further down in the discussion. We hope that by doing so we have also made clearer that these points were both raised together as they are both limitations which may reduce sensitivity, however we can see that this was not very clear in the previous draft. We have also added reference to the asymptomatic population in the penultimate paragraph of the discussion.

11. Pg 13, line 262, Discussion. Further validation of personnel/ occupational data?

Thank you, we have clarified this refers to occupational data.

12. Pg 13, line 262, Discussion. Based on the results (esp the Ct values), one can also infer that LFTs are most useful when pre-test probability is high (e.g. among symptomatic individuals)

We have edited this sentence to reflect this and feel that by mentioning pre-test probability in this section we have been able to further strengthen the point we were making.

---

## [Decision Letter · Decision Letter 1]

8 Aug 2023

Test performance of lateral flow rapid antigen tests for COVID-19 in Welsh adult care home staff using routine surveillance data

PONE-D-23-02428R1

Dear Dr. Hogg,

We’re pleased to inform you that your manuscript has been judged scientifically suitable for publication and will be formally accepted for publication once it meets all outstanding technical requirements.

Kind regards,

André Ricardo Ribas Freitas

Academic Editor

PLOS ONE

Additional Editor Comments (optional):

We appreciate the interest in publishing with PLoS ONE, the manuscript is well suited to the purposes of this journal.

Suggestions made by reviewers were properly responded to and incorporated. As noted by reviewer 2, the only suggestion that can still be adjusted during the next stage of the publication flow is the adequacy of some phrases to improve English that can make the text clearer to readers.

Reviewers' comments:

Reviewer's Responses to Questions

**Comments to the Author**

1. If the authors have adequately addressed your comments raised in a previous round of review and you feel that this manuscript is now acceptable for publication, you may indicate that here to bypass the “Comments to the Author” section, enter your conflict of interest statement in the “Confidential to Editor” section, and submit your "Accept" recommendation.

Reviewer #2: All comments have been addressed

2. Is the manuscript technically sound, and do the data support the conclusions?

Reviewer #2: Yes

3. Has the statistical analysis been performed appropriately and rigorously? 

Reviewer #2: Yes

4. Have the authors made all data underlying the findings in their manuscript fully available?

Reviewer #2: (No Response)

5. Is the manuscript presented in an intelligible fashion and written in standard English?

Reviewer #2: No

6. Review Comments to the Author

Reviewer #2: I thank the authors for addressing my comments in the manuscript. The flow of the manuscript is much improved while keeping the original intent and results presented in the first draft submitted.

Of note, I encourage the authors to do a grammatical review of the discussion section prior to publishing, as there are a few sentences that could be made clearer. Other than that, I enjoyed reading the updated manuscript.

7. PLOS authors have the option to publish the peer review history of their article (what does this mean?). If published, this will include your full peer review and any attached files.

Reviewer #2: No

---

## [Editor Report · Acceptance letter]

14 Aug 2023

PONE-D-23-02428R1 

Test performance of lateral flow rapid antigen tests for COVID-19 in Welsh adult care home staff using routine surveillance data 

Dear Dr. Hogg:

I'm pleased to inform you that your manuscript has been deemed suitable for publication in PLOS ONE. Congratulations! Your manuscript is now with our production department. 

Kind regards, 

on behalf of

Dr. André Ricardo Ribas Freitas 

Academic Editor

PLOS ONE